# Eco-Efficiency and Its Determinants: The Case of the Italian Beef Cattle Sector

Lucio Cecchini [1], Francesco Romagnoli [2], Massimo Chiorri [1,*] and Biancamaria Torquati [1]

[1] Department of Agricultural, Food and Environmental Sciences, University of Perugia, Borgo XX Giugno 74, 06121 Perugia, Italy; lucio.cecchini@unipg.it (L.C.); biancamaria.torquati@unipg.it (B.T.)
[2] Institute of Energy Systems and Environment, Riga Technical University, 12-1 Azenes str., LV-1039 Riga, Latvia; francesco.romagnoli@rtu.lv
* Correspondence: massimo.chiorri@unipg.it; Tel.: +39-075-5856274

**Abstract:** In recent years, eco-efficiency assessment has proven to be an effective tool to reduce the environmental damages of agricultural activities while preserving their economic sustainability. Hence, this paper aims to assess the eco-efficiency of a sample of 148 beef cattle farms operating in the extensive livestock system of Central Italy. The analysis is based on Farm Accountancy Data Network (FADN) economic data in the year 2020 and includes, as environmental pressures, farm expenditure for the use of fuels, electricity and heating, and fertilizers. A two-stage approach was implemented: in the first stage, an input-oriented DEA model including slack variables was used to quantify farm eco-efficiency scores and determine the polluting inputs' abatement potentials. In the second stage, the influence of possible influencing factors on eco-efficiency scores was tested using a regression model for truncated data. The analyzed farms were found to be highly eco-inefficient, as they could abate their environmental pressures, on average, in a range from 56% to 60% while keeping the value of their global production constant. Fertilizers and fuel consumption were identified as the least efficiently operating inputs, with potential reductions in terms of the related expenditures fluctuating between 9% and 42%. Farms showing a high-intensity livestock system, a low labor intensity, and a larger farm area were recognized as the most eco-efficient. Environmental and animal welfare subsidies were found to not affect eco-efficiency, while a negative influence was estimated for a single farm payment, which does not seem to be an incentive mechanism for farms to operate efficiently.

**Keywords:** eco-efficiency; livestock farms; data envelopment analysis; model approaches in estimating greenhouse gases (GHG); truncated regression

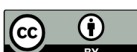

## 1. Introduction

Agricultural activities play a pivotal role in providing access to food, supporting farmers' incomes, and strengthening the resilience of many rural communities, while ensuring a large part of ecosystem services, including biodiversity and landscape conservation.

Nevertheless, this sector is recognized as one of the most responsible for contributing to the deterioration of human and natural ecosystems, as its negative impacts include pollution and degradation of soil, water, and air.

Within the agricultural sector, livestock activities are definitely the ones most responsible for the production of greenhouse gases (GHG), as they contribute significantly to the emissions of carbon dioxide ($CO_2$), methane ($CH_4$), and nitrogen oxide ($N_2O$) [1–3]. In the livestock sector, the largest share of polluting emissions relies on cattle production systems, accounting for 65% [4]. In addition, cattle livestock farming contributes significantly to environmental pollution by affecting water quality and is responsible for more than half of the anthropogenic emissions of N and P [5].

Indeed, the enhancement of the sustainability of agricultural activities represents one of the most crucial challenges identified by the policy agendas. For example, at the European level, the Common Agricultural Policy (CAP), accordingly to the European Green Deal strategy [6], has identified environmental and climate issues as a priority, seeking to enhance the contribution of agriculture to the environmental and climate goals of the European Union (EU), providing higher support to farms adopting improved environmental practices.

In this context, politicians and scientists have increasingly focused their attention on assessing the undesirable outputs of agricultural and livestock systems, with a view to quantifying the related reduction potential and identifying the main critical issues affecting farms' environmental management. In this regard, a large variety of methods, indexes, and measures able to simultaneously consider both the technical efficiency and the environmental performances of the production processes have been proposed. Among these, the assessment of eco-efficiency has been successfully implemented in an increasing number of studies focusing on the agricultural [7–9] and livestock sectors [10–12].

In particular, eco-efficiency assessment is particularly helpful in production contexts, such as livestock farms, where the existence of conflicting economic and environmental goals has been extensively proven [13].

In fact, due to its definition, the eco-efficiency concept is dealing with solutions from a cost-effectiveness perspective, thus allowing the trade-offs between productivity and green practices to be overcome [14]. From this perspective, such an approach seems to be the most suitable to provide win–win solutions in the livestock sector, whose environmental impacts are generally reduced in highly efficient production contexts [15,16]. Overall, two main approaches for the efficiency estimation, calculated as the distance of the Decision Making Units (DMUs) from the theoretically most efficient frontier, were found in the reviewed literature: the parametric one, which entails differences related to the functional form of the underlying distance function, and the non-parametric one, based on Data Envelopment Analysis (DEA) [7].

The first one makes restrictive assumptions about the production function underlying the distance function, thus allowing advantages to be obtained from the computational point of view [17] and from the algebraic manipulation [18]. However, the imposition of a predetermined functional form for the production technology may lead to biases in efficiency score estimates if the technology is mis-specified.

The non-parametric techniques based on the DEA, however, do not provide for strong assumptions concerning the functional form specification, allowing for a greater degree of flexibility, resulting in more consistent and reliable efficiency estimation. Hence, DEA seems to be the most suitable method to be implemented in the case of multiple-input or multiple-output production process efficiency evaluation [19].

Overall, DEA is a methodology based on linear programming techniques that determine the relative efficiency of similar DMUs [20]. The method was first introduced in 1978 by Charnes et al. [21], and its basic formulation assumes a monotonous relationship of linear proportionality between input and output [20], leading to efficiency scores ranging from 0 to 1. The closer the score is to the maximum range, the more efficient the DMU is; the closer it is to the minimum range, the more inefficient the DMU.

However, the monotonicity assumption is violated in the case of the joint production of desired (good) outputs and undesirable (bad) outputs, including, for example, the polluting emissions associated with livestock farming. In fact, an increase in polluting emissions is associated with a decrease in efficiency scores and vice versa.

To take these aspects into account, and based on the growing interest in environmental economics issues, extended DEA models have been developed to model environmental damages able to measure environmental and technical efficiency [22].

According to the existing literature, the strategies for incorporating environmental pollutants can be divided into two main categories: indirect approaches and direct approaches [23,24]. The former envisages that the undesirable outputs would be included in

the traditional CCR (Charnes, Cooper, and Rhodes) and BCC (Banker, Charnes, and Cooper) DEA models, through appropriate transformations. Among these, the inverse additive approach requires that the bad outputs, with a change in sign, were algebraically added to the desired outputs [25]. In another case, bad outputs were considered as inputs in the traditional DEA models [26–28], or, in the inverse multiplicative case, the bad output's reciprocal is treated as good output [29,30].

Several studies focusing on the technical efficiency of livestock farms, based on traditional radial DEA models, have been conducted [31–35]; however, few studies, limited to the dairy sector, have taken into explicit consideration the energy management [36] and the environmental performances [37–40].

Furthermore, indirect approaches are based on radial measures, not allowing the quantification of the excesses at the level of single polluting inputs (input slacks) or the deficiencies in the good output production (output slacks), thus failing to capture the mix-inefficiency [41].

However, adding slack variables to traditional DEA models allows overcoming such limitations, in order to separately discriminate the level of efficiency or inefficiency for each of the output and input items included, as is in the case for the direct approaches.

Many studies focusing on the agricultural sector have recently used DEA models with undesirable outputs, alone or combined in a second-stage framework, at both the national [42–44] and the farm level [9,37,45–47]. In particular, the proposed DEA methodological frameworks have pursued the following two main purposes:

(1) Determination of the eco-efficiency level and the potential output increase (input reduction) in inefficient Decision-Making Units (DMUs), under the hypothesis of different returns to scale;

(2) Analysis of the effects on efficiency scores of structural and environmental factors characterizing farm management.

Although these approaches have been widely adopted to estimate the eco-efficiency of cattle farms [9,10,32–34,48], most of them have focused on the dairy sector and used indirect approaches, without accounting for input and output slacks. Only a few of the existing papers focus on the beef cattle sector [49,50]. In the former, life cycle assessment (LCA) was combined with Slack-Based Measures–DEA to assess the eco-efficiency of 48 Austrian multifunctional farms. However, a limited number of beef-producing cattle farms (8) were taken into consideration; moreover, the use of the LCA impact categories values as DEA inputs does not allow for a direct quantification of the excessive use of polluting inputs (e.g., fertilizers, electricity, fuels, etc.).

The latter implemented a traditional two-stage DEA model, including GHG emissions as a bad output, to assess the effects of EU agri-environmental schemes on the efficiency performances of a sample of French beef cattle farms. In particular, to the best of our knowledge, none of these studies has meant to estimate the eco-efficiency frontier and quantify, at the same time, the potential to reduce GHG emissions and the use of fertilizers. This research attempts to partially fill the gaps found in the existing literature by focusing on the eco-efficiency assessment of a large farm sample representative of the entire central Italy beef cattle sector, with the specific purpose of identifying the main institutional and farm management drivers of eco-efficiency. In this regard, in this paper, a two-stage input-oriented DEA model incorporating slack variables was implemented, based on the Farm Accountancy Data Network (FADN) database, according to the following three objectives: (i) to assess the eco-efficiency of 148 beef cattle farms located in central Italy through Data Envelopment Analysis (DEA), by considering the use of fuels, electricity and heating, and the use of fertilizers as environmental pressures; (ii) to quantify the abatement potential for the considered environmental pressures; (iii) to test for the influence of possible explanatory variables on the eco-efficiency, by implementing a regression model for truncated data, in order to go in depth into the estimated differences in terms of environmental efficiencies between farms.

Hence, our findings attempt to contribute to the growing stream of literature focusing on eco-efficiency assessment in the livestock sector from a twofold perspective. First, our research deepens the understanding of farm eco-efficiency performance by contributing to the debate on the hypothesis that farms showing high level of productivity and technical efficiency are also better performing environmentally. To this end, to provide a comprehensive analysis framework of farms' environmental performance, single pressure-specific eco-efficiency indicators are computed. These results could represent valuable support for producers to improve inappropriate production techniques that may result in excessive input usage and environmental damage.

Secondly, this study is the first that, focusing on beef cattle, seeks to provide a comprehensive methodology framework to evaluate environmental efficiency and identify the related farm-specific driving factors by using the Farm Accountancy Data Network (FADN) data. Thus, this approach could provide useful support for other studies focusing on different production contexts in the view of the EU's environmental performance benchmarking in the beef cattle sector [26]. The paper is structured into four sections: after the introduction, Section 2 presents the adopted methodological framework, going into the depth about the analytical foundations of the implemented two-stage DEA model. The data processes are reported in the same section. The results are provided and discussed in Section 3. In the last part, an overview and concluding remarks are reported, with particular reference to the related policy implications, as well as suggestions for further research.

## 2. Materials and Methods

### 2.1. Data Collection

The final sample population consists of 148 meat-producing cattle farms from Central Italy (Abruzzo, Lazio, Marche and Umbria regions). Farm-level data, referring to the year 2020, were obtained from Farm Accountancy Data Network (FADN), representing the wider EU database concerning the agricultural sector. Hence, the obtained data could be considered representative of the entire cattle livestock sector in central Italy, as a stratified and weighted FADN sample was used. Since the FADN database mainly contains economic and financial data, most of the technical variables included in the analysis were proxied in terms of related expenditures. This approach has already been adopted in several other studies focusing on eco-efficiency assessment [10,11,42,45], for its ease of implementation and ability to provide detailed and reliable information from representative farm samples, besides allowing for comparison between EU countries. In addition, the usage of monetary units avoids possible miscalculations due to the use of physical units, which does not allow accounting for differences in the environmental impacts associated with different inputs (e.g., heating from renewable energies or fossil ones) [51]. In fact, the usage of these inputs, in the same amount, could have the same effect in terms of DEA results.

On the other hand, due to its features, the use of the FADN database entails limitations relating to the availability of some relevant information for the purposes of environmental evaluations, as in this case. For this reason, only GHG emissions that were strictly related to the purchase of inputs were taken into consideration in the analysis. Consequently, emissions coming directly from livestock activities, such as those associated with enteric fermentation, manure management, and disposal, were not considered, nor were the potential emission savings due to the carbon sequestration of grazing and pasture fields.

In this regard, considering a time horizon of one year, the following items have been considered as polluting inputs: fuel costs, electricity, and heating costs as proxies for the contribution in terms of global warming, as well as fertilizer expenditure, which was taken into consideration as a proxy for the use of nutrients. These latter result in environmental damages in terms of water and soil pollution and indirect GHG emissions associated with

the production processes of fertilizers. These same environmental pressures' categories were also considered by [12] and [9] in their eco-efficiency studies, focusing on cattle dairy farms, under the assumption that the higher the expenditures for those inputs, the higher the environmental impacts in terms of contribution to global warming and usage of nutrients.

In addition, livestock units were included in the DEA analysis as technical input. From the output side, the global production value was considered.

### 2.2. Two-Stage DEA Framework

This paper aims to assess the environmental efficiency of beef cattle farms by implementing an input-oriented DEA model, which includes, as technical inputs, undesirable outputs, according to [27].

The choice of the DEA methodology is justified based on the literature discussed in the previous section, which has extensively used this approach to estimate the eco-efficiency frontiers in different agri-food sectors.

In this regard, this study adopted the following two-stage approach [52,53]: in the first stage, the eco-efficiency scores and the abatement potentials of polluting inputs were estimated via CCR and BCC DEA models, including slack variables [27]. As DEA focuses on the efficiency, this approach does not allow for insights about the factors that determine inefficiency; thus, in the second stage, a Tobit regression model [54] for censored data was implemented, including, as a dependent variable, the efficiency scores calculated in the first stage. Structural and management farm parameters related to the farm area, the intensity of the production process, the labor-intensity, and the level of public subsidies are included as covariates, in order to evaluate their influence on eco-efficiency. The deepened understanding of these relationships will provide a concrete and decisive contribution to the growing debate, in terms of both public and scientific discussion, regarding the need to improve the social and economic sustainability of the livestock sector in the European Union.

First Stage: DEA Method

In recent decades, a large stream of DEA models has been proposed, showing differences mainly related to assumptions regarding the functional form, constraints, and nature of the inputs and outputs considered. However, two traditional DEA models have been found to be the most widely adopted in the existing literature, namely (1) the CCR model, which assumes constant returns to scale (CRS), thus providing an evaluation of total eco-efficiency (CRS); (2) the BCC model, which assumes constant returns to scale (CRS), allowing one to estimate the "pure" technical eco-efficiency, as the operating scale factor is not accounted for. Both models can be implemented with their output-oriented or input-oriented version, which are, respectively, aimed at maximizing the output (while keeping constant the input level) or at minimizing the inputs (while keeping the output level). Considering that this paper deals with eco-efficiency estimation with the specific purpose of quantifying the reduction potentials of the considered environmental pressures, the input orientation was chosen and implemented in this study.

Let $k = \{1,2,\ldots,K\}$ represent the DMU set, $I = \{1,2,\ldots,I\}$ the input set, and $j$ the output, where the $k$-th DMU produces $y_{jk}$ units of output by using $x_{ik}$ units of $i$-th inputs. Introducing an intensity variable indicated with $\lambda_k$, the CCR input-oriented model for the DMUo under evaluation can be expressed, in mathematical terms, with the following maximization problem [21]:

$$\operatorname*{Min}_{\theta_o,\lambda_k} \theta_o$$

s.t.

$$\sum_{k=1}^{K} \lambda_k x_{ik} \leq \theta_o x_{io}, \forall i = 1,2, \ldots, I;$$

$$\sum_{k=1}^{K} \lambda_k y_{jk} - y_{jo} \geq 0 \; \forall j = 1;$$

$$\lambda_k \geq 0, k = 1,2,\cdots,K; \tag{1}$$

where $\theta_o^*$ is the optimal value, calculated as the ratio between the input level achievable by DMUo, by keeping constant the actual output level and the actual input level, and $\lambda_k$ is an intensity variable, measuring the extent to which an activity it is used in the production process.

However, model 1 assumes a constant return to scale (CRS), under the hypothesis that all farms would have been operating at their optimal scale; thus, it is not possible for them to obtain any potential eco-efficiency increase by modifying their production scale. However, DMUs often cannot operate at their optimal scale due to several factors such as imperfect market conditions or other internal organization aspects.

To account for this fact, it is possible to relax this hypothesis and estimate the BCC input-oriented model with variable returns to scale (VRS) [55] by adding the convexity constraint (2) to model 1:

$$\sum_{k=1}^{K} \lambda_k = 1; \tag{2}$$

The DMUo reaches radial efficiency when $\theta_o^* = 1, \lambda_o^* = 0$, and $\lambda_k^* = 0 \; \forall k \neq o$, where * indicates the optimal value of each variable. Instead, radial inefficient farms ($\theta_o^* < 1$) may proportionally decrease their use of input, while maintaining their actual output production, by an amount equal to complement to unity of the $\theta_o^*$ value.

However, by adding slack variables to the models specified above, non-radial levels of single-input (or -output) reduction (or increase) that are achievable by those inefficient farms can be estimated. Hence, according to [56], the input slack $s_i^-$ and output slack $s_j^+$ can be estimated by converting inequality constraints to equality ones, as in the following maximization model:

$$\underset{\theta_o,\lambda_k}{\text{Max}} \; \theta_o$$

s.t.

$$\sum_{k=1}^{K} \lambda_k x_{ik} + s_i^- = x_{io}, \forall i = 1,2, \ldots, I;$$

$$\sum_{k=1}^{K} \lambda_k y_{jk} - s_j^+ = \theta_o y_o, \forall j = 1;$$

$$\lambda_k, \geq 0, k = 1,2, \ldots, K$$

$$s_i^- \geq 0, i = 1,2, \ldots, I$$

$$s_j^+ \geq 0, j = 1 \tag{3}$$

where $s_i^-$ and $s_j^+$ represent an excess use of the *i*-th input or a deficiency in the production of the output *j*, respectively.

Whereas the assumption of CRS or VRS for the eco-efficiency assessment is still a matter of debate [45], to account for the effect that the scale to which the DMUs are operating has on their eco-efficiency, both CCR and BCC models were implemented in this study.

The optimal solution to the CCR and BCC model represents, respectively, the technical eco-efficiency (TEE$_o$) and the "pure" technical efficiency (PTEE$_o$) of the DMU$_o$, with PTEE$_o$ and TEE$_o$ varying in the range of [0, 1].

$$\text{SEE}_o = \frac{\text{TEE}_o}{\text{PEE}_o} \tag{4}$$

By taking the ratio between TEE$_o$ and PTEE$_o$, it is possible to compute the scale efficiency, with SEE$_o$ = 1 indicating that the DMU$_o$ operates at the optimal scale, given that the two efficiency scores are equivalent. In the case that SEE$_o$ < 1, the DMU$_o$ operates at increasing returns to scale (IRS), and potential efficiency gains are therefore possible from increasing the production scale; however , SEE$_o$ > 1 indicates that the DMU is operating at decreasing returns to scale (DRS), and thus efficiency improvements would be achievable if DMU reduced its production scale.

### 2.2.2 Estimation Environmental Pressure Abatement Potentials

With respect to the polluting input fuel expenditure, electricity and heating costs, and fertilizers expenditure, in the case of the eco-inefficiency of the DMU$_o$, positive values of the associated slacks variables $s_i^-$ can be estimated, representing the excess use of inputs that the DMU$_o$ could reduce to become eco-efficient, while preserving its global production. Indeed, this value could be also considered as the abatement potential (PA$_o$) of the *i*-th inputs achievable by the DMU:

$$\text{PA}_o = s_i^- \tag{5}$$

Hence, according to Hu and Wang [57], the efficiency of single polluting inputs (IEE$_o$) can be calculated as the ratio between the level of input to be achieved and the current level:

$$\text{IEE}_o = \frac{x_{lo} - s_i^*}{x_{lo}} \tag{6}$$

The IEE$_o$ values range from 0 to 1. The higher the value, the higher the level of efficiency and, consequently, the lower the reduction potential. All the models illustrated above were solved in General Algebraic Modeling System (GAMS) environment.

### 2.2.3. Second Stage: Censored Regression Analysis

To quantify the effect of possible influencing factors on farms' eco-efficiency, a regression model was implemented by considering the scores obtained from the first-stage models as the dependent variable.

Although a variety of econometric modeling strategies have been proposed in this regard, a large stream of literature has shown that the Ordinary Least Squares (OLS) model results in biased and inconsistent parameter estimates when the dependent variable is censored [58]. In addition, the range of the observed dependent variable could be different from that of the predicted eco-efficiency scores. To overcome these limitations, Simar and Wilson [59] proposed a method based on a two-stage double-bootstrapped truncated regression that is able to account for the problems of serial correlations arising when DEA scores are treated as independent observations when they are generated from the same process. However, this approach has been questioned on several grounds, particularly the excessive complexity of its implementation and the lack of robustness compared to other methods [60].

In general, the most common approach used for the second stage is represented by implementing a censored normal regression analysis such as the Tobit model [61–63], assuming censoring at 0 and 1 values. The underlying assumption is that the dependent variables (i.e., $TEE_o$ and $PTEE_o$) are linear, additive, and separable functions of the observed influencing factors.

Hence, the two estimated models could be analytically formalized with the following equation [54]:

$$y_k^* = X_k Z + q + \varepsilon_k \qquad (7)$$

where $y_k^*$ represents the DEA scores obtained from the first stage, observed for values ranging from 0 to 1, and censored otherwise; $X_k$ is the vector of explanatory variables; β is a vector of parameters; q is the unknown intercept; and $\varepsilon_k$~iidN $(0,\sigma^2)$ is the statistical noise.

The maximum likelihood procedure with the White estimator to obtain consistent standard errors was used to implement the two Tobit models, respectively, considering $TEE_o$ and $PTEE_o$ as dependent variables. A backward selection approach was used as an elimination criterion with a *p*-value > 0.1 to remove the insignificant factors from the full model. A partial correlation analysis between the DEA variables and the final models' covariates has been carried out in the pre-estimation phase to obtain unbiased and consistent Tobit estimates [64].

After the estimation, a Link test was carried out to verify the correct model specification [50], and variance inflation factors (VIFs) were used to detect the degree of multicollinearity between regressors.

Given that Z parameters estimates could not be directly interpreted as the marginal effects on the mean value of the observed $y_k$ with respect to a change in $X_k$ [65], two non-linear marginal effects were estimated:

(1) the marginal effects for the expected value of $y_k$ conditional on being uncensored,
(2) the marginal effects for the unconditional expected value of $y_k$ [66]. Such an approach allows us to account for differences, between inefficient and efficient farms, in the effects of the covariates on the efficiency scores. The Stata 12 package was used to carry out all the statistical analyses.

## 3. Results and Discussions

### 3.1. Descriptive Statistics of the First- and Second-Stage Variables

Table 1 shows the descriptive statistics relating to the output and input variables included in the first-stage model.

With respect to the output, an average value of global production equal to 55,639 Euros was estimated for the considered sample; in this regard, a large degree of heterogeneity was found between the sample farms, as evidenced by the width of the min–max range and the high standard deviation values. Focusing on the polluting inputs, the involved farms showed average expenditures of 2512 Euros and 5479 Euros, respectively, for fuels and electricity and heating, with differences related to the size of the herd, the stable management, and crop cultivation techniques. On average, the sample farms spent 1837 Euros for fertilizers. The average herd size was 27.52 Livestock Units (LSUs), varying in a range between 1.3 LSU and 344.2 LSU.

**Table 1.** Descriptive statistics of the input/output variables (*n* = 148).

| Outputs/Inputs | Variable | Unit | Mean | Standard Deviation | Min | Max |
|---|---|---|---|---|---|---|
| Output | Global Production | Euro | 55,639 | 133,138 | 980.00 | 1,138,400 |
| Input | Expenditure on fuels | Euro | 2512 | 6479 | 0.00 | 57,683 |
| Input | Expenditure on electricity and heating | Euro | 5479 | 8021 | 0.00 | 85,142 |
| Input | Expenditure on fertilizers | Euro | 1837 | 2719.22 | 0.00 | 20,186.00 |
| Input | Livestock Unit | LSU | 27.52 | 42.72 | 1.30 | 344.20 |

Table 2 reports the descriptive statistics relating to the variables identified, on the basis of the literature and the availability of data from the FADN database, as eco-efficiency determinants. Specifically, the following farm structural factors were considered according to García-Gudiño et al. [67], Martinsson and Hansson [10], Gomez-Limon et al. [68], and Picazo-Tadeo et al. [45]: farm size, in terms of surface area dedicated to livestock activities; intensity of the livestock production system, in terms of both the value of Global production and labor employment; and the amount of Common Agricultural Policy (CAP) subsidies, by explicitly considering those for animal welfare and environmental aspects.

**Table 2.** Descriptive statistics of the eco-efficiency determinants (*n* = 148).

| | Farm Area | Intensity of Farming | Labor Hours per Livestock Unit | Farm Payment | Animal Welfare Subsidy | Environmental Subsidy |
|---|---|---|---|---|---|---|
| MEAN | 27.66 | 3875 | 82.62 | 7604 | 1280 | 3925 |
| DV. ST. | 42.98 | 11,386 | 131.47 | 10,626 | 4692 | 5560 |
| MIN | 0.30 | 76.92 | 0.56 | 0.00 | 0.00 | 0.00 |
| MAX | 344.20 | 113,032 | 1000 | 107,012 | 33,100 | 56,417 |

*3.2. First Stage: DEA Results*

3.2.1. Eco-Efficiency Estimation

The results from the DEA-BCC and CCR models illustrated above are reported below. Table 3 reports the main summary statistical indices, as well as the relative frequency distributions of farm eco-efficiency (TEE), pure technical eco-efficiency (PTEE), and scale eco-efficiency (SEE), which are also displayed in Figure 1. Focusing on Technical Eco-Efficiency, the sample farms showed a quite low performance level, with an average value of TTE of 0.40, varying from a minimum of 0.05 to a maximum score of 1.

From the frequency distribution, it is possible to notice that the largest part of the farms was located in the lower part of the distribution, with the lowest score range of 0–0.19 being the most numerous class, accounting for 59 farms. Moreover, if the two lower classes are considered together, a TEE-value of <0.4 of two-thirds of the entire sample emerges, highlighting the high level of sample heterogeneity in their eco-efficiency performances. In fact, only 30 farms (19%) exhibit a score greater than 0.6, with the fully technical eco-efficient ones accounting for 7% of the total; the remaining 19 farms were almost equally distributed between the classes, ranging from 0.60 to 0.79 and from 0.80 to 0.99, accounting for 11 and 8 farms, respectively. Focusing on VRS (pure technical eco-efficiency), and thus without considering the effect of the production scale, an higher average PTEE value of 0.44 was estimated. Although one-third of the sample farm is still located in the lowest eco-efficiency class, a translation of the farm distribution towards the upper

classes emerges: 28 farms out of the 148 (18.7%) become fully "pure" technically eco-efficient, with 47 farms having a PTEE greater than 0.6.

These results imply that the livestock farms involved could abate their environmental pressures, on average, by 60% and 56%, respectively, assuming CRS and VRS, while keeping the value of their global production constant.

Our findings are in line with those shown in Grassauer et al. [49], who reported a high level of eco-inefficiency for the Austrian beef cattle sector, as they estimated eco-efficiency scores ranging from 0.17 to 0.44.

Although no other direct comparisons of our results could be made, since no existing studies focusing on the beef cattle sector and implementing the same methodological approach were found, several useful points emerge from the comparison with eco-efficiency analysis concerning the dairy cattle sector.

The involved farms were highly eco-inefficient, exhibiting a potential abatement of environmental pressures significantly higher than those quantified in other previous studies concerning the livestock sector. In particular, by focusing on the German dairy sector, Wettemann and Latacz-Lohmann [33] estimated a GHG abatement potential varying from 4.5% to 11.9%. Similarly, in Iribarren et al. [26], a potential reduction of Global Warming Potential (GWP) of 23% was quantified for Spanish dairy farms, by means of the Slacks-Based Measures–DEA model.

These differences could be attributable to the fact that the dairy sector, compared to the beef-producing one, is a more intensive and technically efficient productive system that is at an advanced stage in the transition towards low-polluting input use. In addition, the two mentioned articles used smaller samples, which could make them appear relatively more efficient for the involved farms [69].

Nevertheless, our results are more consistent with those reported in Urdiales et al. [12] and Picazo-Tadeo et al. [45] and, to an even larger extent, to those estimated in Martinsson and Hansson [10]. In the first two cases, both focusing on the Spanish dairy sector, an average potential radial reduction in several environmental pressures of 36.8% and 44% was found, respectively. Martinsson and Hansson [10], by using FADN data and considering the same environmental pressures as in this study, reported an average eco-efficiency score of 0.29 for Swedish dairy farms, corresponding to a potential reduction in environmental pressures equal to 71%.

From the comparison of the efficiency scores obtained assuming CRS and VRS, it is possible to notice that 15 farms (10%) were operating at the optimal production scale, as a scale efficiency of 1 was estimated. Overall, a relatively high performance was detected for the entire farm sample, since 120 farms (80%) were located in the two upper classes of the SEE distribution (SEE > 0.8), with only 16 farms (11%) exhibiting a SEE score < 0.6.

However, the SEE quantification revealed that 134 farms (90%) showed IRS, thus highlighting how potential eco-efficiency gains from increasing their production scale would be achievable.

In general, SEE was found to be equal to 0.85, on average, varying within a minimum of 0.07 and a maximum of 1.

**Table 3.** Technical, "pure" technical, and scale eco-efficiency (*n* = 148).

| Eco-Efficiency Range | Technical Eco-Efficiency (TEE) | | | "Pure" Technical Eco-Efficiency (PTEE) | | | Scale Eco-Efficiency (SEE) | | |
|---|---|---|---|---|---|---|---|---|---|
| | Mean | *n.* | % | Mean | *n.* | % | Mean | *n.* | % |
| 0.00–0.19 | 0.095 | 59 | 40% | 0.098 | 50 | 34% | 0.111 | 5 | 3% |
| 0.20–0.39 | 0.285 | 40 | 27% | 0.399 | 35 | 23% | 0.271 | 1 | 1% |
| 0.40–0.59 | 0.486 | 20 | 13% | 0.641 | 17 | 11% | 0.444 | 10 | 7% |
| 0.60–0.79 | 0.678 | 11 | 7% | 0.839 | 12 | 8% | 0.680 | 13 | 9% |
| 0.80–0.99 | 0.929 | 8 | 5% | 0.987 | 7 | 5% | 0.929 | 105 | 70% |
| 1 | 1.000 | 11 | 7% | 1.000 | 28 | 19% | 1.000 | 15 | 10% |
| Total | 0.396 | 148 | 100% | 0.439 | 148 | 100% | 0.850 | 148 | 100% |
| No. of farms with Constant Returns to Scale (CRS) | | | | | | | | 15 | 10% |
| **No. of farms with Increasing Returns to Scale (IRS)** | | | | | | | | 133 | 90% |

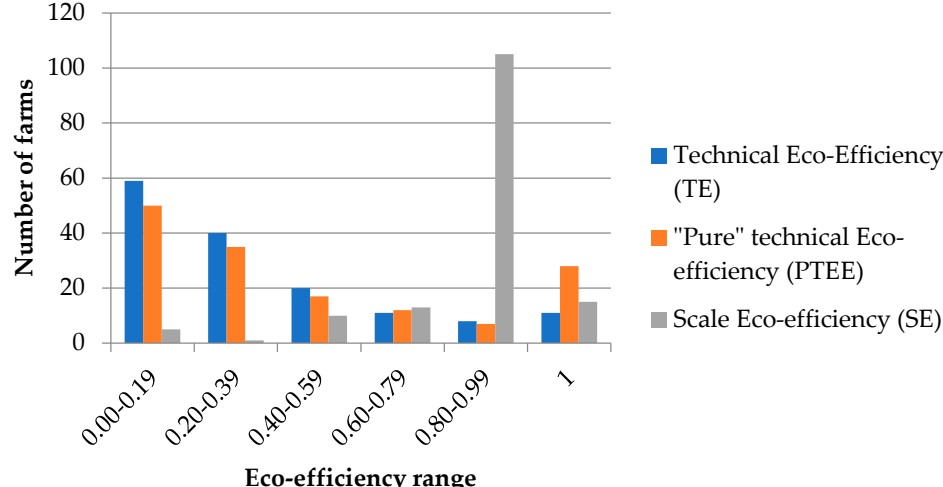

**Figure 1.** Frequency distributions of technical, "pure" technical, and scale eco−efficiency (*n* = 148).

### 3.2.2. Polluting Inputs Potential Reductions

Table 4 shows the most relevant data concerning the estimated input slacks, considering both the CCR and BCC models.

Overall, the number of farms showing CCR input slacks is higher than those estimated from the BCC model, since this latter does not account for the part of eco-inefficiency due to the operating scale. In particular, the largest number of slacks occurred in the case of fuel expenditure (86 farms assuming CRS and 64 farms assuming VRS). On average, these farms could decrease their fuel use by 28,501 (CRS) and 2411 Euros (VRS). In the two assumptions of returns to scale, 63 and 57 farms, respectively were found to have slacks in their expenditure for fertilizers, which could be reduced by 4824 and 3224 Euros, respectively. Concerning electricity and heating expenditure, for approximately 20% of the farms (considering both CRS and VRS), a positive slack was detected, accounting for an average value of 415 and 401, which was quantified considering the entire sample. However, when only the farms with slack are considered, these values increase to 1803 and 1177, respectively. Livestock Units were found to be the input showing the smallest number of farms with a slack (10 assuming CRS), which could decrease their LSU number, on average, by 29 unit. The average value of LSU slack considering the entire sample was 1.95, which is consistent with those estimated by Stępień et al. [9].

Only in two cases was an eco-efficient farm found to have a positive slack, thus revealing that almost all farms with a DEA score equal to one reached strong efficiency.

**Table 4.** Descriptive statistics of input slacks (*n* = 148).

| | Fuels Expenditure | | Fertilizers Expenditure | | Electricity and Heating Expenditure | | Livestock Units | |
|---|---|---|---|---|---|---|---|---|
| | VRS | CRS | VRS | CRS | VRS | CRS | VRS | CRS |
| Farms with slacks | 64 | 86 | 57 | 63 | 27 | 31 | 10 | 0 |
| % of farms with slacks | 43% | 58% | 38% | 42% | 18% | 21% | 7% | 0% |
| Average value of slack (entire sample) | 1753 | 1984 | 2267 | 1794 | 415 | 401 | 1.95 | 0.00 |
| Average value of slack (farms with slack) | 2411 | 2851 | 4824 | 3224 | 1803 | 1177 | 28.95 | 0.00 |
| Full eco-efficient farms with slack | 1 | 0 | 1 | 0 | 0 | 0 | 0 | 0 |
| % of Full eco-efficient farms with slack | 0.67% | 0% | 0.67% | 0% | 0% | 0% | 0% | 0% |

The slacks quantification provides useful insights regarding the main sources of eco-inefficiency, allowing the calculation of single-input eco-efficiency (IEE) (Table 5). In this regard, it is possible to notice that the considered farms perform quite well in terms of LSU (0.97 assuming VRS and 1 assuming CRS) and the use of electricity and heating (0.93 assuming VRS and 0.91 assuming CRS), thus showing that farm size and electricity management do not seem to represent critical points for increasing the sector's competitiveness. Conversely, on average, significantly lower efficiency levels were detected relating to the use of fertilizers (0.82 assuming VRS and 0.80 assuming CRS) and, to a larger extent, to the fuel consumption (0.86 assuming VRS and 0.58 assuming CRS).

Hence, concerning these inputs, the sample farms showed remarkable margins of efficiency improvement, with potential reductions in terms of the related expenditures ranging from 9% to 42%, in the case the inefficient farms would reach level of "fully" efficient ones.

Similarly, Grassauer et al. [49] estimated the highest input slack, among those included in their analysis, for Global Warming Potential (GWP), with related potential abatements ranging from 53% to 94%. These findings provide important and timely support in orienting strategic business choices, both technical and economic, in the medium and long terms. In fact, the exact quantification of the potential of polluting inputs reduction, in terms of the related farm cost savings, is a prime source of information available to livestock management.

**Table 5.** Input eco-efficiency (IEE) of the involved farms (*n* = 148).

| Variable | Min | | Max | | Mean | | Standard Deviation | |
|---|---|---|---|---|---|---|---|---|
| | VRS | CRS | VRS | CRS | VRS | CRS | VRS | CRS |
| Expenditure on fuels | 0.00 | 0.01 | 1.00 | 1 | 0.65 | 0.58 | 0.37 | 0.35 |
| Expenditure on fertilizers | 0.00 | 0.10 | 1.00 | 1 | 0.82 | 0.80 | 0.28 | 0.29 |
| Expenditure on electricity and heating | 0.04 | 0.11 | 1.00 | 1 | 0.93 | 0.91 | 0.21 | 0.21 |
| Livestock Unit | 0.33 | 1.00 | 1.00 | 1 | 0.97 | 1.00 | 0.09 | 0.00 |

In view of deriving some practical indications to help farmers adopt the most appropriate improvement measures, Table 6 reports the input abatement potentials (PA$_o$s)

across five farm size classes. In line with Balezentis et al. [70], the obtained results do not allow clear and precise patterns to be identified for all the considered inputs. Focusing on "pure" technical input management (VRS), regardless of the operational scale, a negative relationship between reduction potential and herd size was detected in regard to fuel consumption. In particular, the highest reduction margin (40.95%) was observed for the farms rearing less than nine LSUs, thus implying that smaller farms could benefit more than larger ones from investment in m machinery. Conversely, further improvements in fertilizers use as well as electricity and heating management should be implemented by farms with more than 100 LSUs, which showed the highest abatement potential among the classes (40.1% and 21.45%, respectively). This class also shows the lowest input efficiency level in terms of LSU, as a 30.31% related abatement potential was estimated. These latter findings highlight how enhancing the productivity level also still represents an open issue also for large-scale operating farms.

**Table 6.** Mean input abatement potentials for farm size classes (%).

| Farm Size Classes (LSU) | Expenditure on Fuels | | Expenditure on Fertilizers | | Expenditure on Electricity and Heating | | Livestock Units | |
|---|---|---|---|---|---|---|---|---|
| | VRS | CRS | VRS | CRS | VRS | CRS | VRS | CRS |
| <9 | 40.95 | 52.65 | 15.40 | 28.62 | 6.78 | 10.41 | 0.00 | 0.00 |
| 9–19.9 | 29.26 | 37.14 | 14.20 | 13.57 | 3.40 | 10.76 | 0.00 | 0.00 |
| 20–49.9 | 30.22 | 32.96 | 19.84 | 13.11 | 6.85 | 7.33 | 1.10 | 0.00 |
| 50–99.9 | 54.96 | 31.90 | 11.10 | 6.70 | 4.55 | 4.63 | 8.86 | 0.00 |
| ≥100 | 18.00 | 37.13 | 40.10 | 24.63 | 21.45 | 0.00 | 30.31 | 0.00 |

### 3.2.3. Second Stage: Eco-Efficiency Determinants

The results from Tobit regression models implemented in the second stage are discussed below. Before considering the model estimates, a brief description of the results obtained from statistical tests carried out in the pre- and post-estimation phases is provided.

A non-collinearity between the covariates was detected from the VIFs test, which resulted in average values of 1.04 and 1.11 when TEE and PTEE were included as dependent variables, respectively. No variable showing VIF > 1.18 was found. No issue of model-uncorrected specification for both the Tobit models was detected by implementing the Link test, as the linear predicted value-squared was found to have no explanatory power.

The estimates from the two backward stepwise Tobit regression models including technical eco-efficiency (TEE), and pure technical eco-efficiency (PTEE) scores as dependent variables are reported in Table 7. The related conditional and unconditional marginal effects are presented in Table 8.

**Table 7.** Tobit regression models estimates.

| Variable | Technical Eco-Efficiency (TEE) | | | Pure Technical Eco-Efficiency (PTEE) | | |
|---|---|---|---|---|---|---|
| | Coef. | | *p*-Value | Coef. | | *p*-Value |
| Intensity of livestock system (Global production per LU) | 0.0034 | *** | 0.000 | 0.0036 | *** | 0.000 |
| Farm payment | −0.00063 | ** | 0.012 | −0.00064 | ** | 0.038 |
| Farm area | 0.024 | *** | 0.000 | 0.036 | *** | 0.000 |

| Labor intensity (Hours per LU) | | | | −0.0012 | *** | 0.006 |
|---|---|---|---|---|---|---|
| _cons | 0.242 | *** | 0.000 | 0.165 | *** | 0.003 |
| Log pseudolikelihood | | −15.362 | | | −57.562 | |
| Number of obs | | 148 | | | 148 | |
| F-statistics | | 12.52 | | | 12.49 | |
| Prob > F | | 0.000 | | | 0.000 | |
| Number of censored observations | | 11 right-censored observations | | | 28 right-censored observations | |
| Pseudo R2 | | 0.7161 | | | 0.4142 | |

\*\* $p < 0.050$, \*\*\* $p < 0.01$.

Overall, the implemented models performed quite well in terms of statistical significance, as four and three of the six considered predictors for the TEE and PTEE models, respectively, were found to significantly affect eco-efficiency, specifically, the intensity of the livestock system, farm payment, farm area, and labor intensity (this latter factor became insignificant when TEE is considered).

**Table 8.** Marginal effects (MEs) of significant variables from the Tobit regression models.

| Variables | Technical Eco-Efficiency (TEE) | | Pure Technical Efficiency (PTEE) | |
|---|---|---|---|---|
| | MEs for the Expected Value of TEE Conditional on Being Uncensored | MEs for the Unconditional Expected Value of TEE | MEs for the Expected Value of PTEE Conditional on Being Uncensored | MEs for the Unconditional Expected Value of TEE |
| Intensity of livestock system (Global production per LU) | 0.00317 | 0.00328 | 0.00537 | 0.00611 |
| Farm payment | −0.00058 | −0.00060 | −0.00050 | −0.00057 |
| Farm area | 0.022 | 0.023 | 0.028 | 0.032 |
| Labour intensity (Hours per LU) | - | - | 0.009 | 0.010 |

However, the effects of the explanatory variables were found to be similar between the two models, implying that scale efficiency does not seem to have any role in influencing the postulated relationship.

As no significant differences between the two estimated marginal effects emerged, the discussion from here on will focus only on the ME for the unconditional expected value.

Our estimates highlight that the intensity of the livestock production system is positively and significantly associated with eco-efficiency, as the associated increases in terms of the unconditional expected values equal 0.36 and 0.34, respectively, for PTEE and TEE models, as pointed out above.

This result is consistent with those estimated by Soteriades et al. [11] and Martinsson and Hansson [10], who, although focusing, respectively, on UK and Swedish cattle dairy sectors, found that an intensive and well-structured production process could positively impact eco-efficiency. This would seem to suggest that policy measures aimed at improving the eco-efficiency of the Italian beef cattle sector should be targeted to increase its productivity performances, which suffer in most cases from a lack of adequate management systems compared with dairy farming.

The estimated negative marginal effects of −0.00060 (TEE) and −0.00057 (PTEE) for farm single payment imply that the greater the subsidy amount, the lower the eco-efficiency score. In addition, both environmental subsidies and animal welfare subsidies seemed to have no significant effects on TEE and PTEE. Until this point, no homogenous results have been reported in the reviewed literature. In particular, in contrast with our study, many authors found that both direct CAP payments and environmental subsidies positively and significantly affect eco-efficiency [45,68,71], as they promote farm investments in better-performing facilities and equipment. However, since these subsides represent direct income support, small and medium-sized farms, which are those mostly characterizing the central Italy beef cattle sector, spent them as current expenditure, with no benefits in terms of environmental performances [9]. In this regard, this type of support does not seem to be an incentive mechanism for farms to improve their production processes, as it promotes the conservation of the status quo concerning farm management, thus promoting investments in other assets [72].

As a matter of fact, many studies have reported a negative relationship between subsidies and farm efficiency, focusing on different research areas and livestock systems [10,73]. This is mainly due to the fact that in many cases, higher direct payments are associated with extensive farms with relatively low productivity and, consequently, low eco-efficiency.

According to other studies, farm area positively influenced farm eco-efficiency, since larger land availability allows farms to more easily afford investments for environmental efficiency improvement and obtain advantages from scale economies.

The estimated marginal effect for variables concerning the labor intensity was significant and negative in the PTEE model, thus confirming that the farms that use less of this input per LSU resulted in higher eco-efficiency scores. This finding is in line with previous studies by [10,74] focusing on dairy farms, remarking that the introduction of labor-saving procedures and technologies could represent an effective strategy to improve both technical and environmental efficiency.

## 4. Conclusions

Eco-efficiency analysis could valuably support the livestock sector in improving its environmental performances, as it could contribute to an effective design of targeted policy measures enhancing farm sustainability. In the present study, a two-stage DEA model, including environmental pressures (GHGs and nutrients) as technical inputs, was implemented to assess the eco-efficiency and identify its main determinants, focusing on a representative sample of 148 beef cattle farms located in central Italy. Our results found that the average eco-efficiency of the involved farms was quite low, ranging from 0.40 to 0.44, when CRS and VRS were assumed, respectively. These results imply that the sample farms could radially abate their environmental pressures, on average, by 60% (CRS) and 56% (VRS), while keeping the value of their global production constant.

To this point, the operation scale does not seem to be a strong source of eco-inefficiency, as only a 15% potential decrease in polluting inputs could be obtained if all the farms were operating at the optimal scale.

However, remarkable differences were found among environmental pressures, as the single polluting inputs' efficiency indicated: farms were found to perform in the most eco-inefficient manner corresponding to the pressures associated with the consumption o fuel and fertilizers, with the potential reduction ranging, respectively, from 35% (VRS) to 42%

(CRS) and from 18% (VRS) to 20% (CRS). In this sense, the exact quantification of the impact-reduction potentials represents valuable information for farmers, who may implement targeted and costless improvement actions in terms of global production value.

The analysis of the eco-efficiency determinants revealed that a high intensity of a livestock system, a low labor intensity, and a larger farm area resulted in increasing eco-efficiency, while an insignificant or negative effect was associated with public subsidies.

Hence, productivity was identified as the main factor affecting eco-inefficiency, thus implying that eco-inefficient farms should focus firstly on improving their management and production processes and optimizing their use of labor and polluting inputs.

In terms of policy implications, these findings highlight the need to provide for specific investment measures aimed at fostering the introduction of technologies and techniques to reduce environmental damages, to be combined with a single payment scheme, rather than provide environmental subsidies based on the farm area.

More specifically, the increase in co-efficiency performances in central Italy may be achieved according to a three-track policy: (i) by promoting support for investment actions and environmentally friendly technologies tailored to raise farms' productivity and improve their input management, from which both economic and environmental performances can benefit; (ii) by improving agro-environmental payment schemes, as they could represent a real incentive for farmers to adopt the best practices observed for the fully technically efficient farms; and (iii) finally, by providing training and advisory programs with the purpose of heightening farmers' awareness of environmental issues, while fostering their knowledge and expertise about production techniques aimed at preventing environmental damages and enhancing animal welfare conditions.

The limitations of using FADN data represent the main weakness of this study. In fact, doing so allows considering only economic production data, with respect to the hypothesis that higher expenditures are associated with higher environmental pressures. However, higher costs could be associated with using inputs with less impact, such as renewable fuels or organic fertilizers, which would result in lower environmental damages. Moreover, the results obtained in terms of eco-efficiency could be related only to that part of pressures associated with the purchased input, as that associated with the in-farm management (i.e., GHG emissions from enteric fermentation or waste disposal) cannot be accounted for. Thus, further research should include technical data involving a wider range of environmental impact categories, as well as test for the influence of other additional eco-efficiency predictors involving management and social aspects. Another limitation is related to the deterministic approach of the DEA framework, which does not allow for interpreting the results in a frame of statistical significance. In this regard, the use of a stochastic framework or bootstrapping techniques may represent possible developments for future studies.

**Author Contributions:** Conceptualization, L.C., M.C., and B.T.; methodology, L.C.; validation, BT, F.R., M.C., and L.C.; formal analysis, L.C.; investigation and data curation M.C.; writing—original draft preparation, L.C.; writing—review and editing, L.C, F.R.; supervision, B.T. F.R.; funding acquisition, B.T. All authors have read and agreed to the published version of the manuscript.

**Funding:** This research was undertaken as part of an Erasmus+ KA2 project titled "Sustainable Well-being Entrepreneurship for Diversification in Agriculture (SWEDA)", which has been funded with support from the European Commission (grant number 2020-1-IT02-KA203-079927, 2020).

**Institutional Review Board Statement:** Not applicable.

**Informed Consent Statement:** Not applicable.

**Data Availability Statement:** The data used for this study were obtained from Council for Agricultural Research and Economics (CREA), but restrictions apply to the availability of these data. The data are available at https://bancadatirica.crea.gov.it/Account/Login.aspx (accessed on 20 July 2022) with permission from CREA.

**Acknowledgments:** We would like to thank Luca Cesaro, Antonio Giampaolo and Luca Turchetti of the "CREA—Research Centre for Agricultural Policies and Bioeconomy" for the valuable information obtained from consulting the Italian Database. We thank the editors and two anonymous reviewers for their insightful and constructive comments and suggestions.

**Conflicts of Interest:** The authors declare no conflicts of interest.

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
