# Peer review of "Eco-Efficiency and Its Determinants: The Case of the Italian Beef Cattle Sector"

_agriculture, doi:10.3390/agriculture13051107_

Round 1

Reviewer 1 Report

The manuscript applied DEA for evaluation the efficiency of beef cattle. The paper is within the scope of Agriculture journal. It an interesting work and can be accepted for publication after a revision. Some of my comments are:

1.     Please revise the whole English language in the paper and review it again

2.     Better discuss on the structure of the model and also compare the results of this paper with other researches

3.     Choice of modeling methods, assumptions and methodologies needs to be suitably justified and/ or provided with apt citations. Also, limitations of the approach should be stated.

4.     More discussion about the novelty of paper.

5.     In the introduction, you need to connect the state of the art to your paper goals. Please follow the literature review by a clear and concise state of the art analysis. This should clearly show the knowledge gaps identified and link them to your paper goals. Please reason both the novelty and the relevance of your paper goals.

6.     The benefits of the results in the wide perspective of industrial production are not discussed i.e. the results should be further elaborated to show how they could be used for the real applications

7.     In the conclusions, in addition to summarizing the actions taken and results, please strengthen the explanation of their significance. It is recommended to use quantitative reasoning comparing with appropriate benchmarks.

8.     Using some of the results in abstract section and improve it.

9.     Add the nomenclature

1Please revise the whole English language in the paper and review it again

Reviewer 2 Report

The paper examined the trade-off between environmental indicators and profitability of a sample of livestock farms in Europe by using DEA methodology. The introduction, data and methodology are appropriate to meet the objectives of the paper.

The paper looks very abstract, author should introduce the concept and theory behind the empirical analysis with some sort of diagrammatic representation how the trade off works in livestock farms. The rationale of choice of the variables needs to be further strengthened. The empirical findings especially distribution of efficiency and slacks may be presented in figures for clarity and appeal.

The sentences are too large through out the paper, many grammatical mistakes are there, please rectify for clarity and readability.

Some examples are given below.

The below sentence in the abstract can be improved ‘The results have shown a low eco-efficient (efficiency) of the analyzed farms, as they could abate their environmental pressures, on average, in a range going from 56% to  60% while keeping the value of their global production constant’

Similarly, in introduction ‘this sector is recognized as one of the most influencing (use right word) the integrity of 33 human and natural ecosystems’ can be improved.

Similar problem is there with the below sentence.

Among these, the assessment of eco-efficiency, meaning the capacity of (to) produce efficiently while using the least possible amount of inputs, has been successfully implemented in an increasing number of studies focusing on the agricultural [8–10] and livestock sector [11–13].

Equations: all symbols are not explained, some problems are there with superscript/subscript.

Authors have not given the distribution of efficiencies by farms; plots of distribution will give good understanding about the firms.

 Technical efficiency is too low less than 20% for more than 50% of the sample, pls check and explain.

Authors can also relate their methodology/conclusions with international literature in agriculture, for reference I suggest Reddy, A. A. (2010). Disparities in agricultural productivity growth in Andhra Pradesh. The Indian Economic Journal58(1), 134-152.

Over all, the paper is interesting and publishable after improvement. 

The sentences are too large through out the paper, many grammatical mistakes are there, please rectify for clarity and readability.

Some examples are given below.

The below sentence in the abstract can be improved ‘The results have shown a low eco-efficient (efficiency) of the analyzed farms, as they could abate their environmental pressures, on average, in a range going from 56% to  60% while keeping the value of their global production constant’

Similarly, in introduction ‘this sector is recognized as one of the most influencing (use right word) the integrity of 33 human and natural ecosystems’ can be improved.

Similar problem is there with the below sentence.

Among these, the assessment of eco-efficiency, meaning the capacity of (to) produce efficiently while using the least possible amount of inputs, has been successfully implemented in an increasing number of studies focusing on the agricultural [8–10] and livestock sector [11–13].

Equations: all symbols are not explained, some problems are there with superscript/subscript.

Reviewer 3 Report

The manuscript “Eco-efficiency and its determinants: the case of Italian beef cattle sector” assesses the eco-efficiency of a sample of 148 beef cattle farms operating in the extensive livestock system of Central Italy, with the non-parametric techniques based on the DEA, and will be interesting to Agriculture readers.

Rich empirical materials (a sample of 148 beef cattle farms), proper research methods, well described, discussion of literature satisfied. The conclusion is interesting.

The authors should be looking for better methods to describe ecoefficiency, but overall, the manuscript is acceptable in its present form to publishing in Agriculture.

Reviewer 4 Report

The paper touches on the current issue of eco-efficiency in relation to performance and subsidies in the beef cattle production sector. As I am not a statistician, I cannot evaluate methodological approaches.

I find the results quite serious, especially the small relation between subsidies and eco-efficiency - although the discussion is necessarily limited by the database available (FADN). Tables 1 and 2 indicate significant differences between the studied farms, which may also be relevant for the interpretation of the results.

The readability of the article is limited by the large number of abbreviations, which are explained, but the readers should get a dictionary if they do not want to keep looking for the meanings of individual abbreviations. GHGs in keywords must be given in full. I recommend keeping the abbreviations where they appear as statistical variables, but using the full names where the interpretation of the results is concerned. Also, presenting absolute values, which are usually in the thousands, to two decimal places, impairs the readability of the document. In the subsection Data collection, the first part is repeated.

Overall, I consider the article successful and beneficial. I recommend its publication (after removing some minor flaws).

Round 2

Reviewer 1 Report

Dear authors

I think the present format of this paper can be published and no more edition is needed. Congratulations.

Reviewer 2 Report

The authors addressed most of the earlier comments, now it is ready for the publication.

The authors addressed most of the earlier comments, now it is ready for the publication.